# Outcomes of COVID-19 patients with comorbidities in southwest Nigeria

**Akin Osibogun**[1,2]*, **Mobolanle Balogun**[2], **Akin Abayomi**[1,3], **Jide Idris**[1], **Yetunde Kuyinu**[4], **Oluwakemi Odukoya**[2], **Ololade Wright**[4], **Remi Adeseun**[1], **Bamidele Mutiu**[5], **Babatunde Saka**[5], **Nike Osa**[1], **Dayo Lajide**[1], **Ismael Abdus-Salam**[1], **Bodunrin Osikomaiya**[6], **Oluwatosin Onasanya**[7], **Bisola Adebayo**[4], **Yewande Oshodi**[8], **Sunday Adesola**[9], **Olu Adejumo**[9], **Olufemi Erinoso**[10], **Hussein Abdur-Razzaq**[3], **Abimbola Bowale**[9], **Kingsley Akinroye**[11]

**1** Lagos State COVID-19 Incident Command System/Emergency Operation Centre, Lagos, Nigeria, **2** Department of Community Health and Primary Care, College of Medicine University of Lagos, Lagos, Nigeria, **3** Lagos State Ministry of Health, Lagos, Nigeria, **4** Department of Community Health and Primary Health Care, Lagos State University College of Medicine, Lagos, Nigeria, **5** Lagos State Biosafety and Biosecurity Governing Council, Lagos, Nigeria, **6** Lagos State Blood Transfusion Service, Lagos State Ministry of Health, Lagos, Nigeria, **7** Lagos State Primary Health Care Board, Lagos, Nigeria, **8** Department of Psychiatry, College of Medicine University of Lagos, Lagos, Nigeria, **9** Mainland Hospital, Yaba, Lagos, Nigeria, **10** Department of Oral and Maxillofacial Surgery, Lagos State University Teaching Hospital, Ikeja, Nigeria, **11** Nigerian Heart Foundation, Lagos, Nigeria

* akinosibogun@yahoo.co.uk

**Data Availability Statement:** All relevant data are available within the paper and its Supporting Information File.

**Funding:** This study was funded by the Lagos State Government as part of its COVID-19 Pandemic

## Abstract

### Background

Data on the comorbidities that result in negative outcomes for people with COVID-19 are currently scarce for African populations. This study identifies comorbidities that predict death among a large sample of COVID-19 patients from Nigeria.

### Methods

This was a retrospective analysis of medical records for 2184 laboratory confirmed cases of COVID-19 in Lagos, southwest Nigeria. Extracted data included age, sex, severity of condition at presentation and self-reported comorbidities. The outcomes of interest were death or discharge from facility.

### Results

Most of the cases were male (65.8%) and the median age was 43 years (IQR: 33–55). Four hundred and ninety-two patients (22.5%) had at least one comorbidity and the most common amongst them were hypertension (74.2%) and diabetes (30.3%). The mortality rate was 3.3% and a significantly higher proportion of patients with comorbidities died compared to those with none. The comorbidities that predicted death were hypertension (OR: 2.21, 95%CI: 1.22–4.01), diabetes (OR: 3.69, 95% CI: 1.99–6.85), renal disease (OR: 12.53, 95%CI: 1.97–79.56), cancer (OR: 14.12, 95% CI: 2.03–98.19) and HIV (OR: 1.77–84.15].

countermeasure. The State Government was however not involved in the design and implementation of the study, nor with its data analysis and manuscript writing. All views expressed in the manuscript are those of the authors only.

**Competing interests:** The authors have declared that no competing interests exist.

## Conclusion

Comorbidities are prevalent and the associated risk of death is high among COVID-19 patients in Lagos, Nigeria. Public enlightenment, early identification and targeted care for COVID-19 cases with comorbidities are recommended as the pandemic evolves.

## Introduction

Nigeria is among the top 5 countries in sub-Saharan Africa (SSA) with the highest numbers of confirmed cases of COVID-19 [1]. The first case of COVID-19 in Nigeria, which was an imported case from Italy was confirmed on the 27th of February 2020 in Lagos, southwest Nigeria [2]. Lagos, being a major economic center and the most populous state in Nigeria has since become the epicenter of COVID-19 in Nigeria with established community transmission accounting for 15, 414 (34.7%) of the 44, 433 confirmed cases and 192 (21.1%) of the 910 reported deaths in Nigeria as at August 5, 2020 [3, 4].

In response to the outbreak, the Lagos state government in collaboration with the private sector in the state established ten dedicated isolation and treatment centers for COVID-19 patients [5]. Most cases that have been admitted have been mild to moderate since the early stages of the outbreak [6]. However, there has been an increase in the number of severe cases and deaths as a result of COVID-19, which makes it imperative to evaluate the risk factors for progression of the disease.

Current evidence from China and the US suggests that comorbidities such as hypertension, diabetes, obesity, chronic obstructive pulmonary disease (COPD) and cerebrovascular disease increase the risk of severity and death from COVID-19 [7–10]. However, there are significant differences in demographic patterns and disease trends between high-income and low- and middle-income countries [11] and limited data on implicating comorbidities from the latter.

Recently, in Lagos state, there was a policy shift to home-based care for COVID-19 patients with mild disease [12]. To further guide control efforts, it is important to accurately identify groups at risk so as to inform risk communication to communities and early management of COVID-19. This study therefore aims to identify the comorbidities that are associated with death in the Nigerian population using a large sample of confirmed patients admitted in care in the nation's epicenter as case study.

## Methods

The reporting of this study was guided by the STROBE statement for reporting observational studies [13] (S1 Checklist).

### Study design, setting and participants

This was a retrospective study, which used medical records of 2469 suspected and confirmed COVID-19 patients. The cases reported in this study were patients treated in ten isolation and treatment facilities for COVID-19 across Lagos, southwest Nigeria. These facilities had a combined bed capacity of 674. Each facility had holding spaces for suspected undiagnosed patients (contacts of positive cases or with symptoms suggestive of COVID-19) pending confirmation of results. Those with negative results were discharged from the holding spaces while all those that were confirmed positive were admitted for care. The strategy at the phase of the outbreak

during the study period was containment with an attempt to get in all positive cases of COVID-19 for isolation in order to reduce community transmission.

We used medical records of COVID-19 patients admitted for care between February 27, 2020 and July 6, 2020. Inclusion criteria was laboratory confirmed cases with naso-oropharyngeal and sputum specimens that tested positive for the SARS-CoV-2 virus by using real-time reverse-transcription–polymerase-chain-reaction (RT-PCR) assay for SARS-CoV-2 in accordance with the protocol established by WHO [14]. Those that were suspected cases without confirmation of a positive result were excluded from the final analysis.

## Data extraction and description of variables

Data was extracted from electronic medical records of the cases as maintained by the central case management pillar of the emergency response team for Lagos state. We obtained data on patients' age, sex, severity of condition upon presentation at the facilities, self-reported comorbidities and outcomes of interest i.e. death and recovery (evidenced by discharge) as at the study end date of July 6, 2020. Patients who were still in care or were evacuated or transferred out by the end date were classified as having an undetermined outcome.

Severity of condition of patients upon presentation at the hospital was based on clinical symptoms and the need for oxygen and ventilation. A patient that was asymptomatic at presentation was classified as mild while a patient was classified as moderate if they presented with fever, cough, respiratory rate <30 breaths per minutes and peripheral capillary oxygen saturation (spO2) >90% for adults and >92% for children. A patient with grunting respiration, respiratory rate >30 breaths per minute, spO2 <90% for adults and <92% for children requiring oxygenation was classified as severe while a patient with respiratory failure requiring mechanical ventilation was classified as critical. Our case definition was adapted from a handbook on clinical experience in China [6, 15, 16].

Discharge of patients was based on a negative PCR-based SARS-CoV-2 virus test. Comorbidities were majorly presented as recorded while those with frequencies less than 5 were regrouped and classified according to organ systems.

## Data analysis

Data analysis was done using STATA version 15 (StataCorp, USA). Continuous variables were tested for the assumption of normality using histograms and the Shapiro Wilk test. Categorical variables were presented in frequencies and percentages, normally distributed continuous variables were presented as mean and standard deviation (SD) while non-normal continuous variables were presented as median and interquartile range (IQR). Chi-square and Wilcoxon rank sum tests were used to determine differences in patient characteristics between those with comorbidities and those without. To determine comorbidity risk factors for death, we used unadjusted odds ratios (OR) and excluded comorbidities with no outcome of death. The remaining comorbidities were imputed into logistic regression models to determine those that are predictors of death. This was done irrespective of statistical significance in bivariate analyses as they were considered a priori to affect the probability of death. Age and sex of patients were considered as confounding variables in the final model. Adjusted odds ratios (AOR) and 95% confidence intervals (CI) were computed for each predictor variable. Goodness of fit was evaluated using the Hosmer-Lemeshow test. The results were assessed to be significant at $p < 0.05$.

## Ethics

Ethical approval was obtained from the Health Research and Ethics Committee of the Lagos State University Teaching Hospital. To ensure confidentiality, patient data (date range:

February 27, 2020 to July 6, 2020) was fully anonymized before being accessed, saved on pass-worded computers and handled only by authorized personnel. Informed consent was not required for this study as it depended on previously collected service data and a waiver was issued by the ethics committee.

## Results

Out of 2469 patients admitted into the facilities within the study period, 2184 (88.46%) were laboratory-confirmed COVID-19 cases. Subsequent analysis was done among the confirmed cases in line with the eligibility criteria. Most of the confirmed COVID-19 cases in this study were male (65.8%) and in the age group of 1 to 40 years (44%). They ranged in age from 0 to 98 years with a median age of 43 years (IQR: 33–55). Over half (57.5%) of them were admitted into the facilities in mild clinical condition and 73 of them (3.3%) died while 1725 (79%) were discharged during the period of study. Four hundred and ninety-two patients (22.5%) had at least one co-morbidity related to COVID-19 and the median number of morbidities was 1 (IQR: 1–2). Compared to patients without comorbidities, significantly higher proportions of patients with comorbidities were aged 50 years and above, presented in severe to critical condition and died while on admission (p<0.05) (Table 1).

Among 492 patients with comorbidities, the most common ones were hypertension (74.2%), diabetes (30.3%) and asthma (10.2%). More male than female patients with

**Table 1. Profile of 2184 confirmed COVID-19 patients with or without comorbidities admitted in Lagos isolation centers from January 27 to July 6, 2020.**

| Variables | All patients | Patients with comorbidity | Patients with no comorbidity | Chi-square | p-value |
|---|---|---|---|---|---|
| | N = 2184 | N = 492 | N = 1692 | | |
| **Age group (years)** | | | | 432.87 | **<0.001** |
| <40 | 956(44.95) | 62(12.60) | 894(53.12) | | |
| 40–49 | 530(24.37) | 100(20.33) | 430(25.55) | | |
| 50–59 | 316(14.53) | 119(24.19) | 197(11.71) | | |
| ≥ 60 | 373(17.15) | 211(42.89) | 162(9.63) | | |
| Total | 2175[a] | 492 | 1683 | | |
| **Median age (IQR)** | 43(33–55) | 57 (47–67) | 39 (30–48) | | **<0.001** |
| **Sex** | | | | 0.99 | 0.320 |
| Male | 1436(65.81) | 333(67.68) | 1103(65.27) | | |
| Female | 746(34.19) | 159(32.32) | 587(34.73) | | |
| Total | 2182[b] | 492 | 1690 | | |
| **Severity of condition at presentation** | | | | 216.63 | **<0.001** |
| Mild | 1251(57.65) | 181 (36.79) | 1070(63.77) | | |
| Moderate | 770(35.48) | 214(43.50) | 556(33.13) | | |
| Severe | 107(4.93) | 65(13.21) | 42(2.50) | | |
| Critical | 42(1.94) | 32(6.50) | 10(0.60) | | |
| Total | 2170[c] | 492 | 1678 | | |
| **Outcomes** | | | | 107.74 | **<0.001** |
| Died | 73(3.34) | 51(10.37) | 22(1.30) | | |
| Discharged | 1725(78.98) | 336(68.29) | 1389(82.09) | | |
| Undetermined[d] | 386 (17.67) | 105(21.34) | 281(16.61) | | |
| Total | 2184 | 492 | 1692 | | |

[a]Missing: 9 (0.41%), [b]Missing: 2 (0.09%),

[c]Missing: 14 (0.64%)

[d]Still in care: 358 (16.39%); Evacuated/transferred out: 28 (1.18%)

comorbidities had hypertension and diabetes. Higher proportions of patients with 2 or more comorbidities were aged 60 years and above (56.4%), male (73%) and presented in severe (16.7%) and critical (13.5%) conditions (Fig 1).

The comorbidities that were identified as risk factors for death were hypertension (OR: 7.36; 95%CI: 4.55–11.89), diabetes (OR: 10.67; 95%CI: 6.31–18.07), renal disease (OR: 33.28; 95%CI: 7.31–151.56), cancer (OR: 9.69; 95%CI: 1.85–50.81), cardiovascular disease (OR: 6.91; 95%CI: 1.41–33.49) and HIV (OR: 9.69; 95%CI: 1.85–50.81). Patients with two or more of these comorbidities were about four times more likely to die than those with one comorbidity. Other identified risk factors for death were older age groups (50–59 years and ≥ 60 years), male sex and presentation in moderate to critical conditions (Table 2).

Table 3 shows the multiple logistic regression examining the comorbidities that predict death. In model one, having hypertension, diabetes, renal disease, cancer, and HIV predicted death from COVID-19. In model two, after controlling for age and sex, these five comorbidities remained as significant predictors of death from COVID-19. Compared to patients without these comorbidities, patients with hypertension were 2.21 times more likely to die from COVID-19 (95% CI: 1.22–4.01), patients with diabetes were 3.69 times more likely to die (95% CI: 1.99–6.85), those with renal disease were 12.53 times more likely to die (95% CI: 1.97–79.56), those with cancer were 14.12 times more likely to die (95% CI: 2.03–98.19) while those with HIV were 12.21 times were more likely to die from COVID-19 (95% CI: 1.77–84.15]. In addition, male patients were 1.84 times more likely to die than the female (95% CI: 1.01–3.39); also, patients between 50 to 59 years were 3 times more likely (95% CI: 1.23–7.32) and those aged 60 years and above were 6.87 times more likely (95% CI: 2.98–15.85) to die from COVID-19 compared to patients less than 40 years.

## Discussion

This study is presently one of the few to determine risks of comorbid conditions among a large sample of COVID-19 patients in Nigeria and in Africa. In this study, we discovered that the predominant comorbidities were hypertension and diabetes and that patients with comorbidities were more likely to die from COVID-19, more so when they had 2 or more comorbidities. We also found that the comorbidities that predicted death were hypertension, diabetes, renal disease, cancer and HIV.

The mortality rate within our cohort (3.3%) was higher than the national rate of 2%, possibly because our study included only hospital admissions and because Lagos state has the highest of burden of disease in the country. Higher mortality rates have been reported in China, Europe and the US while most countries in Africa have had lower mortality than the global trend [10, 17–19]. While the reasons for this is not fully understood, some hypotheses postulated include the warmer African weather and youthful population [19].

Similar to our findings, hypertension and diabetes were among the most common comorbidities in China and the US [7–10]. The epidemiological transition in sub-Saharan Africa has resulted in a rise of non-communicable diseases in the presence of longstanding burden of infectious diseases [20]. The burden of hypertension and diabetes in Nigeria is high with national prevalence of 28.9% and 5.77% respectively [21, 22]. There are still a lot of undiagnosed cases as a result of lack of awareness and poor health seeking behavior of the population [22, 23]. As a result, many patients present with uncontrolled diseases with attending complications [24]. Importantly, hypertension and diabetes are known to co-exist in patients and our findings indicate that having multiple comorbidities increases the risk of death from COVID-19 similar to a previous study in China [7].

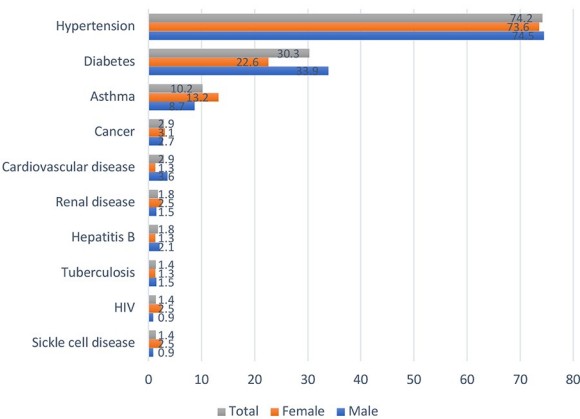

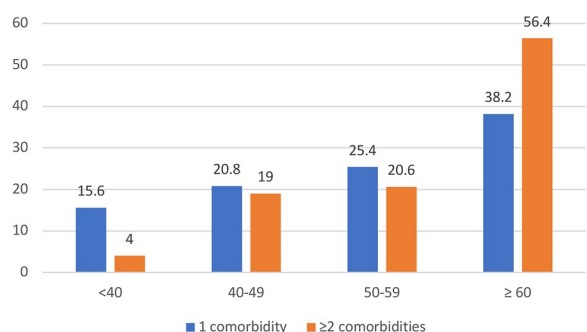

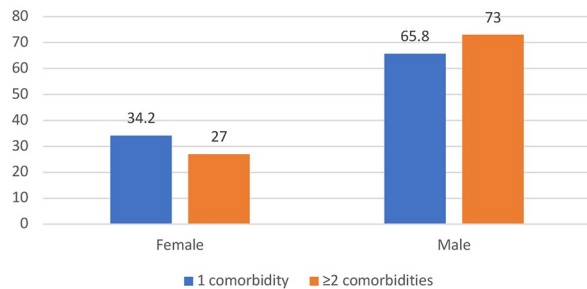

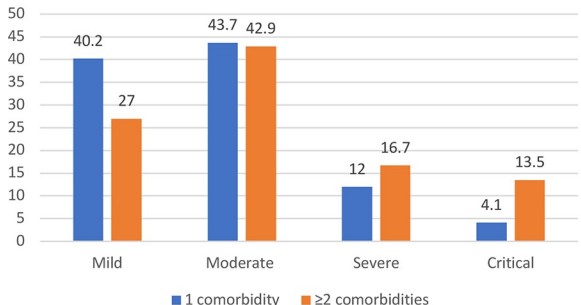

**Fig 1. Types and number of comorbidities among 492 COVID-19 patients with comorbidities admitted in Lagos isolation centers from January 27 to July 6, 2020.**

Unlike previous studies in developed countries [7–10], COPD and cerebrovascular diseases were not reported by patients in this study. However, this may not imply that they are not prevalent in Nigeria. COPD, for example, is thought to be prevalent in Africa but largely undiagnosed [25]. Certainly, the risk factors such as tobacco smoking, biomass smoke exposure, poverty, poor nutrition, pulmonary tuberculosis and HIV infection are prevalent in Nigeria. Similarly, stroke is prevalent in Africa and occurs frequently as a complication of hypertension [26].

Regarding comorbidities that predicted death, our study echoes previous ones in finding hypertension and diabetes as risk factors for mortality [7–9, 27]. We also identified that renal disease and cancer predicted death from COVID-19 with the odds being 13 and 14 times higher than those without these conditions respectively. Although, the stage of disease could not be deduced from secondary data, late-stage presentation for renal disease and cancer have been documented among Nigerians because of lack of awareness, fear, misconceptions, misdiagnosis, lengthy investigation time and reliance on unorthodox treatment [28, 29].

**Table 2. Bivariate analyses of COVID-19 comorbidities as risk factors for death.**

| Variables | Died | Discharged | OR [95%CI] | p-value |
|---|---|---|---|---|
| | N = 73 | N = 1725 | | |
| **Comorbidities*** | | | | |
| Hypertension | 40(14.08) | 244(85.92) | 7.36[4.55–11.89] | <**0.001** |
| Diabetes | 26(23.42) | 85(76.58) | 10.67[6.31–18.07] | <**0.001** |
| Asthma | 3(6.67) | 42(93.33) | 1.72[0.52–5.68] | 0.375 |
| Renal disease | 4(57.14) | 3(42.86) | 33.28[7.31–151.56] | <**0.001** |
| Cancer | 2(28.57) | 5(71.43) | 9.69[1.85–50.81] | **0.007** |
| Cardiovascular disease | 2(22.22) | 12(77.78) | 6.91[1.41–33.88] | **0.017** |
| Hepatitis B | 1(14.29) | 6(85.71) | 3.98[0.47–33.49] | 0.204 |
| HIV | 2(28.57) | 5(71.43) | 9.69[1.85–50.81] | **0.007** |
| **Number of comorbidities** | N = 51 | N = 336 | | |
| 1 | 25(8.47) | 270(91.53) | Ref. | |
| ≥ 2 | 26(28.26) | 66(71.74) | 4.25[2.31–7.84] | <**0.001** |
| **Age group (years)** | | | | |
| <40 | 9(1.12) | 791(98.88) | Ref. | |
| 40–49 | 5(1.10) | 451(98.90) | 0.97[0.32–2.93] | 0.963 |
| 50–59 | 15(5.79) | 244(94.21) | 5.40[2.34–12.50] | <**0.001** |
| ≥ 60 | 44(15.55) | 239(84.45) | 16.18[7.79–33.63] | <**0.001** |
| **Sex** | | | | |
| Female | 17(2.71) | 611(97.29) | Ref. | |
| Male | 56(4.79) | 1114(95.21) | 1.81[1.04–3.14] | **0.036** |
| **Severity at presentation** | | | | |
| Mild | 4(0.37) | 1089(99.63) | Ref. | |
| Moderate | 16(2.67) | 584(97.33) | 7.46[2.48–22.41] | <**0.001** |
| Severe | 16(23.53) | 52(76.47) | 83.77[27.05–259.43] | <**0.001** |
| Critical | 37(100) | 0(0.0) | 1 | |

OR: Unadjusted odds ratio, Ref: Reference category

*TB and Sickle cell disease excluded because of collinearity: there was no occurrence of death among the patients with both conditions

**Table 3. Multiple logistic regression of COVID-19 comorbidities that predict death.**

| Variables | Model 1 (N = 1798) | Model 2* (N = 2175) |
|---|---|---|
| | AOR [95%CI]; p-value | AOR [95%CI]; p-value |
| **Asthma** | 1.52[0.41–5.57]; 0.529 | 1.81[0.49–6.65]; 0.374 |
| **Hypertension** | 4.45[2.57–7.72]; <0.001 | 2.21 [1.22–4.01]; 0.009 |
| **Diabetes** | 5.31[2.90–9.76]; <0.001 | 3.69[1.99–6.85]; <0.001 |
| **Renal disease** | 17.48[3.05–100.16]; 0.001 | 12.53 [1.97–79.56]; 0.007 |
| **Cancer** | 13.89[2.29–84.33]; 0.004 | 14.12[2.03–98.19]; 0.007 |
| **Cardiovascular disease** | 0.87[0.09–8.10]; 0.905 | 0.64[0.07–5.42]; 0.679 |
| **HIV** | 16.13[2.74–95.07]; 0.002 | 12.21[1.77–84.15]; 0.011 |
| **Hepatitis B** | 1.70[0.15–19.34]; 0.668 | 4.06[0.39–42.26]; 0.241 |
| **Age group (years)** | | |
| <40 | | Ref. |
| 40–49 | | 0.68[0.22–2.09]; 0.496 |
| 50–59 | | 3.00[1.23–7.32]; 0.015 |
| ≥ 60 | | 6.87[2.98–15.85]; <0.001 |
| **Sex** | | |
| Female | | Ref. |
| Male | | 1.84[1.01–3.39]; 0.048 |

* Hosmer-Lemeshow test of goodness of fit p-value: 0.508

Late presentation coupled with dearth of medical personnel, diagnostic and therapeutic equipment, medication and prohibitive cost of treatment, most of which is out of pocket, implies that patients with these diseases may already have a poor prognosis even without COVID-19 [30–33].

Patients with self-reported HIV also had high odds of dying from COVID-19 in our study. Previous studies have shown mixed outcomes among people with HIV and COVID-19. Positive outcomes have been reported in case series from Italy [34], Spain [35], Germany [36] and the US [37]. However, a recent systematic review reported high proportion of death among COVID-19 and HIV co-infected patients [38] and HIV doubled the risk of COVID-19 mortality in South Africa [39]. Nigeria has one of the largest numbers of people living with HIV in SSA and there is greater concern for those with low CD4 cell count, advanced disease, high viral load, and those not on antiretroviral treatment (ART) given that COVID-19 also has adverse immunological and clinical effects [36, 38]. We were however not privy to information on treatment and viral suppression of HIV-infected patients in this study.

Additional findings in our study showed that male patients were twice as likely to die from COVID-19 than the female patients and risk of death was higher among patients that were 50 years and older with that risk being higher among older adults i.e. 60 years and over. The higher risk of death in men and older adults is similar to global trends [40, 41]. Apart from the higher contribution of men to comorbidities, other suggested reasons for their higher morbidity and mortality include behavioral, social and biological differences that favor women [40]. In this study however, the occurrence of comorbidities was not significantly higher in male patients.

Certain limitations must be considered in our study. First, documented comorbidities were based on self-reporting which could result in underestimation of the true burden of comorbid conditions in the patients since some may be undiagnosed. Secondly, both morbidity and mortality from COVID-19 could be underestimated because only hospitalized patients were used.

Thirdly, we used secondary data which had some missing data and did not document some potentially confounding data such as staging, control and treatment of comorbid conditions. Finally, some patients' outcomes were not determined within our study timeframe, which if known, might have provided different assessments of study outcomes than represented in this study.

In conclusion, this study describes the outcomes of patients with comorbidities in large sample of COVID-19 cases. As the COVID-19 pandemic evolves, it is crucial to understand risk factors for disease progression and our study helps to fill gaps that currently exist for African populations. It is recommended that public enlightenment should be emphatic about high-risk comorbidities and that identification through screening be done to identify undiagnosed cases. Persons with comorbidities should be encouraged to use all recommended protective measures [42] and seek prompt care for COVID-19. Furthermore, specific guidelines should be developed for home-based and hospital management of COVID-19 patients with comorbidities especially in the African context so that targeted care can be provided to minimize deaths. More research of a prospective nature is required to provide further evidence of the association between comorbidities and COVID-19 in Africa.

## Supporting information

**S1 Checklist. STROBE statement checklist.**
(PDF)

**S1 Dataset. Dataset for 2184 confirmed COVID-19 patients.**
(XLSX)

## Acknowledgments

The authors wish to place on record the full support of the Incident Commander for the response to the COVID-19 outbreak in Lagos State, Mr Babajide Sanwo-Olu, Governor of Lagos State and the Deputy Governor Dr Kadiri Hamza by making available resources to conduct research as part of the counter-measures to the outbreak. Several health workers in the frontline were responsible for caring for the patients and keeping the records.

## Author Contributions

**Conceptualization:** Akin Osibogun, Akin Abayomi, Jide Idris, Ololade Wright, Remi Adeseun, Kingsley Akinroye.

**Data curation:** Mobolanle Balogun, Bamidele Mutiu, Babatunde Saka, Oluwatosin Onasanya, Bisola Adebayo, Abimbola Bowale.

**Formal analysis:** Akin Osibogun, Mobolanle Balogun, Oluwakemi Odukoya, Ololade Wright.

**Funding acquisition:** Akin Abayomi, Dayo Lajide, Hussein Abdur-Razzaq.

**Methodology:** Akin Osibogun, Akin Abayomi, Jide Idris, Yetunde Kuyinu, Remi Adeseun.

**Project administration:** Akin Osibogun, Nike Osa, Dayo Lajide, Ismael Abdus-Salam, Sunday Adesola, Olu Adejumo, Olufemi Erinoso, Hussein Abdur-Razzaq, Abimbola Bowale.

**Resources:** Remi Adeseun, Hussein Abdur-Razzaq.

**Supervision:** Bodunrin Osikomaiya, Oluwatosin Onasanya, Bisola Adebayo, Yewande Oshodi, Sunday Adesola, Olu Adejumo, Olufemi Erinoso, Abimbola Bowale.

**Writing – original draft:** Mobolanle Balogun.

**Writing – review & editing:** Akin Osibogun, Akin Abayomi, Jide Idris, Yetunde Kuyinu, Olu-
wakemi Odukoya, Ololade Wright, Bamidele Mutiu, Babatunde Saka, Nike Osa, Dayo
Lajide, Ismael Abdus-Salam, Bodunrin Osikomaiya, Oluwatosin Onasanya, Bisola Ade-
bayo, Yewande Oshodi, Sunday Adesola, Olu Adejumo, Olufemi Erinoso, Hussein Abdur-
Razzaq, Abimbola Bowale, Kingsley Akinroye.

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
