## [Decision Letter · Decision Letter 0]

4 Jan 2021

PONE-D-20-28457

Outcomes of COVID-19 patients with comorbidities in southwest Nigeria

PLOS ONE

Dear Dr. Osibogun,

Thank you for submitting your manuscript to PLOS ONE. After careful consideration, we feel that it has merit but does not fully meet PLOS ONE’s publication criteria as it currently stands. Therefore, we invite you to submit a revised version of the manuscript that addresses the points raised during the review process.

We look forward to receiving your revised manuscript.

Kind regards,

Bolajoko O. Olusanya, MBBS, FMCPaed, FRCPCH, PhD

Academic Editor

PLOS ONE

Journal Requirements:

2. In the ethics statement in the manuscript and in the online submission form, please provide additional information about the patient records used in your retrospective study, including: a) whether all data were fully anonymized before you accessed them and/or whether the IRB or ethics committee waived the requirement for informed consent; b) the date range (month and year) during which patients' medical records were accessed.

3.We note that you have indicated that data from this study are available upon request. PLOS only allows data to be available upon request if there are legal or ethical restrictions on sharing data publicly. For information on unacceptable data access restrictions, please see http://journals.plos.org/plosone/s/data-availability#loc-unacceptable-data-access-restrictions.

Reviewers' comments:

Reviewer's Responses to Questions

**Comments to the Author**

1. Is the manuscript technically sound, and do the data support the conclusions?

Reviewer #1: Yes

Reviewer #2: Yes

2. Has the statistical analysis been performed appropriately and rigorously? 

Reviewer #1: Yes

Reviewer #2: Yes

3. Have the authors made all data underlying the findings in their manuscript fully available?

Reviewer #1: No

Reviewer #2: No

4. Is the manuscript presented in an intelligible fashion and written in standard English?

Reviewer #1: Yes

Reviewer #2: Yes

5. Review Comments to the Author

Reviewer #1: The article is interesting ans is well-written. Conclusions are based on the results and responded to the objective of the study.

Please, specify if you followed the STROBE guidelines for retrospective studies. It is not mentioned in the text.

Reviewer #2: Dear author and editor

The manuscript entitled “Outcomes of COVID-19 patients with comorbidities in southwest Nigeria” is an interesting work that adds evidence to the fact that comorbidities are a risk factor for developing complications and evolving to death across different populations, and in this particular case in a cohort of patients in Africa. The manuscript is well written and presents important and interesting results. However, I have some important comments that I consider should be addressed:

Major comments:

Methods:

1. Please adhere to the STROBE statement for reporting of observational studies, and state this accordingly in the manuscript.

Results:

1. This work is apparently related to this published paper: 10.1016/j.ijid.2020.10.024 . Since some of the results are related I recommend citing them properly.

2. Data presented in Figure 1 can be easily summarized in a couple of lines in the text. In order to preserve this Figure, I will recommend making this figure a composite figure with, for example, Panel A showing the frequency of each comorbidity grouped by sex, Panel B the number of comorbidities by age group, and Panel C number of comorbidities and severity at presentation.

Discussion:

1. The authors claimed: “This study is the first to determine risks of comorbid conditions among a large sample of COVID-19 patients in Nigeria and one of the first in Africa”. However, other studies have been published in this respect, particularly: PMID: 33296477. I suggest editing this claim accordingly.

2. The authors claimed that they discovered “that the predominant comorbidities were non-communicable diseases”. However, according to what is stated in the methods, other communicable diseases different than COVID-19 were not assessed in the cohort so making it difficult to sustain this claim. I would suggest to consider this particular issue as a limitation in the study and edit the claim accordingly.

6. PLOS authors have the option to publish the peer review history of their article (what does this mean?). If published, this will include your full peer review and any attached files.

Reviewer #1: No

Reviewer #2: No

---

## [Author Response · Author response to Decision Letter 0]

27 Jan 2021

JOURNAL REQUIREMENTS

Response: We have ensured that the manuscript from the title page to the references meets the style requirements of the journal.

2. In the ethics statement in the manuscript and in the online submission form, please provide additional information about the patient records used in your retrospective study, including: a) whether all data were fully anonymized before you accessed them and/or whether the IRB or ethics committee waived the requirement for informed consent; b) the date range (month and year) during which patients' medical records were accessed.

Response: We have provided the additional information in both the manuscript and the online submission form. 

3.We note that you have indicated that data from this study are available upon request. PLOS only allows data to be available upon request if there are legal or ethical restrictions on sharing data publicly… If there are no restrictions, please upload the minimal anonymized data set necessary to replicate your study findings as either Supporting Information files or to a stable, public repository and provide us with the relevant URLs, DOIs, or accession numbers.

Response: We have included the minimal anonymized dataset as a Supporting Information file

REVIEWER 1

The article is interesting ans is well-written. Conclusions are based on the results and responded to the objective of the study.

Please, specify if you followed the STROBE guidelines for retrospective studies. It is not mentioned in the text.

Response: Thank you for your kind comment. We have followed the STROBE statement for reporting observational studies and indicated so in the first sentence under Methods.

REVIEWER 2

Comment #1: 

The manuscript entitled “Outcomes of COVID-19 patients with comorbidities in southwest Nigeria” is an interesting work that adds evidence to the fact that comorbidities are a risk factor for developing complications and evolving to death across different populations, and in this particular case in a cohort of patients in Africa. The manuscript is well written and presents important and interesting results.

Response: Thank you for your kind comments.

Comment #2: 

Methods:

1. Please adhere to the STROBE statement for reporting of observational studies, and state this accordingly in the manuscript.

Response: We have followed the STROBE statement for reporting observational studies and indicated so in the first sentence under Methods.

Comment #3: 

Results:

1. This work is apparently related to this published paper: 10.1016/j.ijid.2020.10.024 . Since some of the results are related I recommend citing them properly.

Response: Thank you for picking that. The analyses and write up for this current study and the one cited above were done at the same time, which was why it was not cited originally. However, we have gone ahead to cite it in the methods – reference number 16.

Comment #4: 

2. Data presented in Figure 1 can be easily summarized in a couple of lines in the text. In order to preserve this Figure, I will recommend making this figure a composite figure with, for example, Panel A showing the frequency of each comorbidity grouped by sex, Panel B the number of comorbidities by age group, and Panel C number of comorbidities and severity at presentation.

Response: We have gone ahead to make Figures 1a – 1d which can be combined as a composite Figure 1. The figures depict comorbidities grouped by sex, number of comorbidities by age, by sex and by severity at presentation. Additional report has been included in the results to reflect this. 

Comment #5: 

Discussion:

1. The authors claimed: “This study is the first to determine risks of comorbid conditions among a large sample of COVID-19 patients in Nigeria and one of the first in Africa”. However, other studies have been published in this respect, particularly: PMID: 33296477. I suggest editing this claim accordingly.

Response: Thank you for picking that. We have edited the claim to read “This study is presently one of the few to determine risks of comorbid conditions among a large sample of COVID-19 patients in Nigeria and in Africa” 

Comment #6: 

2. The authors claimed that they discovered “that the predominant comorbidities were non-communicable diseases”. However, according to what is stated in the methods, other communicable diseases different than COVID-19 were not assessed in the cohort so making it difficult to sustain this claim. I would suggest to consider this particular issue as a limitation in the study and edit the claim accordingly.

Response: We have edited the claim to read “that the predominant comorbidities were hypertension and diabetes”. However, since we assessed communicable diseases namely tuberculosis, HIV and hepatitis B among the cohort, we decided not to consider this as a limitation in the study. Thank you.

---

## [Decision Letter · Decision Letter 1]

24 Feb 2021

Outcomes of COVID-19 Patients with Comorbidities in Southwest Nigeria

PONE-D-20-28457R1

Dear Dr. Osibogun,

We’re pleased to inform you that your manuscript has been judged scientifically suitable for publication and will be formally accepted for publication once it meets all outstanding technical requirements.

Kind regards,

Bolajoko O. Olusanya, MBBS, FMCPaed, FRCPCH, PhD

Academic Editor

PLOS ONE

Additional Editor Comments (optional):

Authors should introduce a STROBE Statement checklist as a supplementary Table.

Reviewers' comments:

Reviewer's Responses to Questions

**Comments to the Author**

1. If the authors have adequately addressed your comments raised in a previous round of review and you feel that this manuscript is now acceptable for publication, you may indicate that here to bypass the “Comments to the Author” section, enter your conflict of interest statement in the “Confidential to Editor” section, and submit your "Accept" recommendation.

Reviewer #2: All comments have been addressed

2. Is the manuscript technically sound, and do the data support the conclusions?

Reviewer #2: Yes

3. Has the statistical analysis been performed appropriately and rigorously? 

Reviewer #2: Yes

4. Have the authors made all data underlying the findings in their manuscript fully available?

Reviewer #2: Yes

5. Is the manuscript presented in an intelligible fashion and written in standard English?

Reviewer #2: Yes

6. Review Comments to the Author

Reviewer #2: Dear Author and Editor,

I have completed my revision of the manuscript entitled “Outcomes of COVID-19 Patients with Comorbidities in Southwest Nigeria”. It is a very interesting manuscript that highlights the evidence of comorbidities as a risk factor for complications and death across different populations with COVID-19. My previous comments have been properly addressed.

7. PLOS authors have the option to publish the peer review history of their article (what does this mean?). If published, this will include your full peer review and any attached files.

Reviewer #2: No

---

## [Editor Report · Acceptance letter]

1 Mar 2021

PONE-D-20-28457R1 

Outcomes of COVID-19 patients with comorbidities in southwest Nigeria 

Dear Dr. Osibogun:

I'm pleased to inform you that your manuscript has been deemed suitable for publication in PLOS ONE. Congratulations! Your manuscript is now with our production department. 

Kind regards, 

on behalf of

Dr. Bolajoko O. Olusanya 

Academic Editor

PLOS ONE